# Evaluation of Antibacterial Drugs Using Silkworms Infected by *Cutibacterium acnes*

**DOI:** 10.3390/insects12070619

**Published:** 2021-07-08

**Authors:** Yasuhiko Matsumoto, Yuki Tateyama, Takashi Sugita

**Affiliations:** Department of Microbiology, Meiji Pharmaceutical University, 2-522-1 Noshio, Kiyose 204-8588, Tokyo, Japan; m196219@alm.my-pharm.ac.jp (Y.T.); sugita@my-pharm.ac.jp (T.S.)

**Keywords:** silkworm, *Cutibacterium acnes*, infection, antibacterial drugs

## Abstract

**Simple Summary:**

*Cutibacterium acnes*, a common bacterium on human skin, is a causative agent of inflammatory skin diseases and systemic infections. Systemic infections caused by *C. acnes* are difficult to treat because of the drug resistance to typical macrolides. The development of a systemic infection model for *C. acnes* contributes to searching candidates for drug discovery. Silkworm is a useful model animal for evaluating the efficacy of compounds. Several silkworm infection models have been established for the identification of virulence-related genes and novel antibacterial drugs. However, the systemic infection model of *C. acnes* using silkworms was not yet established. We established a new silkworm infection model with *C. acnes* and evaluated the efficacy of antibacterial drugs using the silkworm infection model. The silkworm infection model might be used to identify drug candidates for the treatment against *C. acnes* infection.

**Abstract:**

*Cutibacterium acnes* is a causative agent of inflammatory skin diseases and systemic infections. Systemic infections caused by *C. acnes* are difficult to treat, and the development of a systemic infection model for *C. acnes* would be useful for elucidating the mechanisms of infection and searching for therapeutic agents. In this study, we established a silkworm infection model as a new experimental system to evaluate the interaction between *C. acnes* and the host, and the efficacy of antibacterial drugs. Silkworms infected with *C. acnes* died when reared at 37 °C. The dose of injected bacterial cells required to kill half of the silkworms (LD_50_) was determined under rearing conditions at 37 °C. The viable cell number of *C. acnes* was increased in the hemolymph and fat body of the infected silkworms. Silkworms injected with autoclaved *C. acnes* cells did not die during the study period. The survival time of silkworms injected with *C. acnes* was prolonged by the injection of antibacterial drugs such as tetracycline and clindamycin. These findings suggest that the silkworm *C. acnes* infection model can be used to evaluate host toxicity caused by *C. acnes* and the *in vivo* efficacy of antimicrobial drugs.

## 1. Introduction

*Cutibacterium acnes* (formerly *Propionibacterium acnes*), a common bacterium on human skin, causes inflammatory skin diseases and systemic infections [1,2]. *C. acnes* is isolated as the predominant species in 34% [3] or 36.2% [4] of intervertebral discs removed from patients with chronic low back pain, such as disc herniation. Biofilm formation by *C. acnes* on implants and intervertebral discs causes bloodstream infections [5,6,7]. Because *C. acnes* forms a biofilm and more than 50% of clinically isolated *C. acnes* are resistant to typical macrolides, systemic infections caused by *C. acnes* are difficult to treat [8,9]. Therefore, the development of treatments for systemic infections caused by *C. acnes* is clinically important. Although mammalian animal models of *C. acnes* infection have been established, their use for the evaluation of antibacterial drugs is difficult due to the long duration of the infection [10,11]. Infection experiments using a large number of mammalian animals are also difficult to perform due to animal welfare issues [12].

Silkworms are useful animals for assessing host–pathogen interactions in systemic infections and for evaluating the therapeutic effects of antimicrobial drugs [12,13,14]. Because silkworms also have advantageous features, such as easy rearing in large numbers in a small space with few ethical issues, experiments with large numbers of silkworms can be performed [15]. Moreover, quantitative drug administration and monitoring of parameters in silkworm blood can be performed due to the ease of sample injection and blood collection [12,16,17]. The use of a silkworm infection model based on these advantageous features led to the discovery of virulence-related genes for pathogenic microorganisms such as *Staphylococcus aureus*, *Candida albicans*, and *Candida glabrata* [18,19,20,21]. Further, exploratory studies of antimicrobial drugs using a silkworm infection model led to the identification of compounds exhibiting therapeutic efficacy in mouse infection experiments [22,23,24]. Therefore, the silkworm infection model is useful for studies aimed at elucidating the infection mechanisms of pathogenic microorganisms and evaluating the efficacy of antimicrobial drugs.

In the present study, we attempted to establish an animal model of systemic infection by *C. acnes* using silkworms. We found that injection of *C. acnes* cells killed silkworms. Survival times of the infected silkworms were prolonged by injection of tetracycline and clindamycin. Our findings suggest that the silkworm infection model with *C. acnes* is useful for evaluating the efficacy of antimicrobial drugs.

## 2. Materials and Methods

### 2.1. Reagents

Tetracycline was purchased from FUJIFILM Wako Pure Chemical Corporation (Osaka, Japan). Clindamycin was purchased from Tokyo Chemical Industry Co., Ltd. (Tokyo, Japan). These reagents were dissolved in physiologic saline solution (0.9% NaCl). GAM agar was purchased from Nissui Pharmaceutical Co., Ltd. (Tokyo, Japan).

### 2.2. Culture of C. acnes

*C. acnes* ATCC6919 strain was used in this study. The *C. acnes* ATCC6919 strain was grown on GAM agar plates at 37 °C for 3 days under anaerobic conditions. Physiologic saline (2 mL) was added to the agar plate and suspended using a cell spreader. The *C. acnes* cell suspension was collected from the agar plate and measured absorbance at 600 nm using a spectrophotometer (U-1100; Hitachi Ltd., Tokyo, Japan). The *C. acnes* cells were suspended in physiologic saline to 5–10 of absorbance at 600 nm and the cell suspension was used in the experiments. For calculating the cell number in the cell suspension, the suspension of *C. acnes* cells was 10^7^-fold diluted with saline, and a 100 μL aliquot was spread on a GAM agar plate. After incubation at 37 °C for 3 days under anaerobic conditions, the number of colonies was counted for determining the colony-forming unit (CFU).

### 2.3. Silkworm Rearing

The silkworm rearing was performed as previously described [25]. Eggs of silkworms (Hu Yo × Tukuba Ne) were purchased from Ehime-Sanshu Co., Ltd. (Ehime, Japan), disinfected, and hatched at 25–27 °C. The silkworms were fed an artificial diet, Silkmate 2S, containing antibiotics purchased from Ehime-Sanshu Co., Ltd. (Ehime, Japan). Fifth instar larvae were used in the infection experiments.

### 2.4. Silkworm Infection Experiments

The silkworm infection experiments were performed as previously described [25]. Silkworm fifth instar larvae were fed an artificial diet (1.5 g; Silkmate 2S; Ehime-Sanshu Co., Ltd., Ehime, Japan) overnight. A suspension (50 μL) of the *C. acnes* cells was injected into the silkworm hemolymph with a 1 mL tuberculin syringe (Terumo Medical Corporation, Tokyo, Japan). Silkworms injected with the *C. acnes* cells were placed in an incubator and survival was monitored.

### 2.5. Measurement of Viable Cell Number of C. acnes in Silkworms

Silkworms were injected with *C. acnes* cell suspension (3.3 × 10^8^ cells/50 µL) and incubated at 37 °C. Hemolymph and fat body were harvested from the silkworms at 1 or 24 h after injection. Hemolymph was collected from the larva through a cut on the first proleg according to the previous report [16]. For calculating the *C. acnes* cell number in the hemolymph, the hemolymph was 10^3^- or 10^4^-fold diluted with saline, and a 100 μL aliquot was spread on GAM agar plates. The number of colonies was counted after incubation at 37 °C for 3 days under anaerobic conditions. The viable cell number of *C. acnes* in the hemolymph was calculated as the colony-forming unit (CFU) per 1 mL.

Fat body isolation was performed as previously described [26]. Silkworms were placed on ice for 15 min. The fat body was isolated from the dorsolateral region of each silkworm and rinsed in saline. The wet weight of the fat body was measured. The fat body was homogenized in 1 mL of saline, and the lysate was obtained. The lysate was 10^2^ or 10^3^-diluted with saline, and a 100 μL aliquot was spread on GAM agar plates. The number of colonies was counted after incubation at 37 °C for 3 days under anaerobic conditions. The viable cell number of *C. acnes* per wet weight of the fat body was determined.

### 2.6. LD_50_ Measurement

LD_50_ values, which is the dose of *C. acnes* required to kill half of the silkworms, were determined according to a previous report with slight modification [27]. The *C. acnes* ATCC6919 strain was grown on GAM agar plates at 37 °C under anaerobic conditions for 3 days. *C. acnes* cells grown on GAM agar plate were suspended with 2 mL of physiologic saline. A 2- or 4-fold dilution series of the bacterial suspension was prepared. The *C. acnes* cell suspension (Experiment 1: 2.5 × 10^7^–1.6 × 10^9^ cells/50 µL, Experiment 2: 2.1 × 10^7^–1.3 × 10^9^ cells/50 µL, or Experiment 3: 7.8 × 10^5^–4.1 × 10^8^ cells/50 µL) was injected into the silkworm hemolymph and incubated at 37 °C. Survival of the silkworms at 48 h was monitored. The LD_50_ was determined by a nonlinear regression model using Prism 9 (GraphPad Software, LLC, San Diego, CA, USA, https://www.graphpad.com/scientific-software/prism/, accessed on 25 April 2021).

### 2.7. Evaluation of Therapeutic Activities of Antibacterial Drugs Using Silkworms

Therapeutic activity tests using silkworms were performed according to a previous report with slight modification [25]. Saline (50 μL) or *C. acnes* suspension (3.6 × 10^8^ cells/50 µL) was injected into the hemolymph of silkworms. Immediately after inoculation with the *C. acnes* suspension, 50 µL of an antibacterial drug (50 µg/g larva or 10 µg/g larva) was injected into the hemolymph of the silkworms.

### 2.8. Statistical Analysis

The significance of differences between groups on survival time of silkworms was calculated using a log-rank test based on the Kaplan–Meier method using Prism 9 (GraphPad Software, LLC, San Diego, CA, USA, https://www.graphpad.com/scientific-software/prism/, accessed on 25 April 2021). The statistically significant differences between groups on viable cell number of *C. acnes* in silkworm body were evaluated using the Student *t*-test. *p* < 0.05 was considered a statistically significant difference. Raw data are shown in Appendix A.

## 3. Results

### 3.1. Pathogenicity of C. acnes against Silkworms

Silkworm models of infection by various microorganisms have been established [15]. The body temperature of silkworms, which can be regulated by changing the rearing temperature, is important for bacterial pathogenicity against silkworms [15]. In the case of *Aspergillus fumigatus* and dermatophytes, silkworms do not die at 37 °C by depending on their infection [24,28]. Therefore, it is important to consider the temperature at which the experiment is performed for each pathogen. *C. acnes* grows at approximately 32 °C on the skin and 37 °C in the human body. We examined the rearing temperatures that caused death in *C. acnes*-infected silkworms. Silkworms injected with *C. acnes* (1.6 × 10^9^ cells) reared at 37 °C after injection died within 40 h, whereas infected silkworms reared at 32 °C survived longer (Figure 1). The time required for half of the infected silkworms to die (LT_50_) was 48 h for silkworms reared at 32 °C and 27 h for those reared at 37 °C (Figure 1). The LD_50_ value, which is the bacterial number required to kill half of the silkworms, was 1–3 × 10^8^ cells when infected silkworms were reared at 37 °C (Figure 2). These results suggest that rearing *C. acnes*-infected silkworms at 37 °C decreased survival and that the pathogenicity of *C. acnes* can be quantitatively assessed based on the LD_50_ value.

### 3.2. Increase in Viable Cell Number of C. acnes in Silkworms

We next investigated whether *C. acnes* cells can grow in the silkworm body. The scheme of the experiment is shown in Figure 3a. The viable cell number of *C. acnes* was increased in hemolymph, the blood of silkworm (Figure 3b). Silkworm has a fat body, which is a soft organ that has a similar function to the liver and adipose tissue in mammals. The viable cell number of *C. acnes* was also increased in the fat body of silkworms (Figure 3c). The result suggests that *C. acnes* can proliferate in the silkworm body.

### 3.3. Effect of Heat-Killed C. acnes Cells on Silkworms

Bacterial components such as peptidoglycans of *Porphyromonas gingivalis* lead to shock in silkworms, resulting in their death [29]. Injection of viable or heat-killed *P. gingivalis* cells causes silkworm death [29]. Under conditions in which shock occurs, silkworms cannot be treated with antibiotics [29]. Therefore, it is important to determine whether the death of silkworms by injection of a pathogen is shock by the bacterial components. We next examined the heat-killed cells to evaluate whether *C. acnes* cells must be alive to exert pathogenicity against silkworms. Injection of *C. acnes* cells killed silkworms, but injection of autoclaved *C. acnes* cells did not (Figure 4). These results suggest that the heat-stable components of *C. acnes* do not cause acute silkworm death in the experimental condition.

### 3.4. Therapeutic Effects of Antibacterial Drugs against Silkworms Infected with C. acnes

Tetracycline and clindamycin are used for the treatment of dermatologic disorders including acne vulgaris and are often administrated orally [30,31,32]. We next examined the efficacy of these antibacterial drugs in the silkworm *C. acnes* infection model. In the clinic, tetracycline and clindamycin are administered 250–500 mg/person/day and 600 mg/person/day, respectively [30,31,32]. Based on the information, we determined injection amounts of the antibacterial drugs in the experiments. Administration of tetracycline and clindamycin to silkworms infected with *C. acnes* prolonged the survival time (Figure 5). These results suggest that the efficacy of antibacterial drugs can be evaluated using the silkworm infection model with *C. acnes*.

## 4. Discussion

In this study, we demonstrated that *C. acnes* kills silkworms reared at 37 °C and that the silkworm infection model can be used to evaluate the efficacy of antibacterial drugs. Our findings suggest that the silkworm infection model is useful for assessing pathogenicity and the efficacy of antimicrobial drugs against systemic infection by *C. acnes*.

Because *C. acnes* causes bloodstream infections, we selected injection the *C. acnes* into the hemolymph of the silkworms. *C. acnes*-infected silkworms under the rearing condition at 37 °C were more sensitive than that at 32 °C. We assumed that this is due to two factors: the effect of high-temperature stress on the silkworm and the optimal temperature at which the pathogens have high virulence. The silkworms injected with the autoclaved *C. acnes* cells seem to be unhealthy compared with those in the saline group (Figure 4b). We assumed the reason is that the hemolymph of silkworms was melanized and the melanization might affect the silkworm condition. The *C. acnes* cells grow in the hemolymph and on the tissues of the silkworms. *C. acnes* use the nutrition of the silkworms and might cause toxicity to the tissues. The melanization of the hemolymph of silkworms caused by *C. acnes* might affect tissue toxicity. We assumed that the silkworms infected with *C. acnes* cause sepsis by the proliferation of *C. acnes* cells in the hemolymph and on tissues of the silkworms.

*C. acnes* ATCC6919 used in this study is a type strain whose genome is publicly available and can invade osteoclasts and osteoblasts [33]. Therefore, we used the *C. acnes* ATCC6919 for systemic infection experiments using silkworms. The viable cell number of *C. acnes* was increased in the hemolymph and fat body of silkworms. The results suggest that *C. acnes* causes bacteremia by growth in the silkworm body. *C. acnes* forms a biofilm on the intervertebral discs [5]. We assumed that *C. acnes* might form a biofilm on the fat body of silkworms. Future work will be to evaluate the difference in pathogenicity and drug-resistant in clinical isolates from patients with disc herniation. Moreover, identification of high pathogenic *C. acnes* clinical strain using the silkworm infection model and genetic study of the high pathogenic strain and drug-resistant strain will be an important subject.

*C. acnes* causes superficial inflammatory diseases and systemic infections. In this study, we focused on the systemic infections caused by *C. acnes* and established the systemic infection model using silkworms. We assumed that the silkworm *C. acnes* infection model established in this study is not a superficial inflammatory disease model. The establishment of a superficial inflammatory disease model using silkworms is a future subject for the evaluation of topical drugs.

In a previous study using nucleus pulposus-derived disc cells in vitro, *C. acnes* cells (10^7^–10^8^ cells/mL) were co-cultured with the disc cells and proinflammatory responses of the disc cells were induced by the addition of the *C. acnes* cells [34]. We injected *C. acnes* cells (3.3 × 10^8^ cells/larva), and the range of viable cell number of *C. acnes* in silkworm hemolymph at 1 h after the injection was 3.2–4.8 × 10^7^ cells/mL (Figure 3). Moreover, the LD_50_ was 1–3 × 10^8^ cells when infected silkworms were reared at 37 °C (Figure 2). Therefore, we used *C. acnes* cells (10^8^–10^9^ cells/larva) in this study, because the amounts are needed for silkworm death and the similar range of *C. acnes* cells in previous *in vitro* study.

In the silkworm *C. acnes* infection model established in this study, administration of heat-killed bacteria did not kill the silkworm. Further, *C. acnes*-infected silkworms could be effectively treated with antibacterial drugs, suggesting that the growth of *C. acnes* in the body of silkworms is important for its pathogenicity. In the clinic, tetracycline and clindamycin are administered 250–500 mg/person/day and 600 mg/person/day, respectively [30,31,32]. Given that the average weight of humans is 60 kg, the calculated values of tetracycline and clindamycin are 4.2–8.3 and 10 mg/kg, respectively, for humans. The bodyweight of the silkworm used in this study was approximately 2 g. One-hundred micrograms (50 µg/g larva) and 20 µg (10 µg/g larva) of tetracycline and clindamycin were administered to each silkworm (Figure 5). Therefore, doses of tetracycline and clindamycin were 50 mg/kg or 10 mg/kg. The 10 mg/kg dose is a similar dose to human clinical use. The results support that the silkworm infection model is useful for the evaluation of antibacterial drugs against systemic infection by *C. acnes*. Silkworm infection models are useful for identifying virulence factors of pathogenic microorganisms [13,15]. Further studies are needed to determine which factors in *C. acnes* are responsible for the pathogenicity against silkworms. 

The pharmacokinetics of antimicrobial agents are similar between silkworms and mammals, and antimicrobial drug efficacy in the silkworm infection model can be evaluated based on the pharmacokinetics [23,35,36,37]. A previous study demonstrated that total body clearance (CL_tot_), distribution volume in steady state (Vd_ss_), and the elimination half-life (*t*_1/2_) of antibiotics in silkworm were calculated [37]. For example, CL_tot_, Vd_ss_, and *t*_1/2_ of tetracycline in silkworm were 0.07 mL/g/h, 0.6 mL/g, and 5.5h, respectively. In rabbit, these values of tetracycline were 0.35 mL/g/h, 1.0 mL/g, and 2.0 h, respectively. The pharmacokinetic parameters of tetracycline in silkworms and rabbits are not different at least 10-fold [37]. Moreover, silkworm infection models are useful to develop for in vivo screening of new antimicrobial agents [22,24]. The silkworm *C. acnes* infection model may be useful for identifying effective antibacterial compounds against systemic infection by *C. acnes*. Azole antifungals exhibit antimicrobial activity against *C. acnes* in vitro, and ketoconazole inhibits the lipase activity of *C. acnes* [38,39]. Further studies are needed to identify effective compounds against systemic *C. acnes* infections from among clinically applied drugs using the silkworm infection model.

Recently, an infection model with *C. acnes* using a nematode, *Caenorhabditis elegans*, was reported [40]. *C. elegans* is useful for identifying host factors against *C. acnes* infection based on genetic approaches [40]. The differences between the silkworm system and the *C. elegans* system are that the silkworm blood can be directly injected with *C. acnes* and its pathogenicity at 37 °C, the same temperature as the human body, can be verified. *C. elegans* is difficult to inject quantitatively into body fluids and cannot grow at 37 °C [41]. The silkworm infection model might allow us to identify virulence factors of *C. acnes* at the body temperature of humans.

## 5. Conclusions

We established a silkworm infection model with *C. acnes* and found the system to be useful for evaluating antibacterial drug efficacy. Further studies are needed to determine the clinical applicability of research using the silkworm *C. acnes* infection model.

## Figures and Tables

**Figure 1 insects-12-00619-f001:**
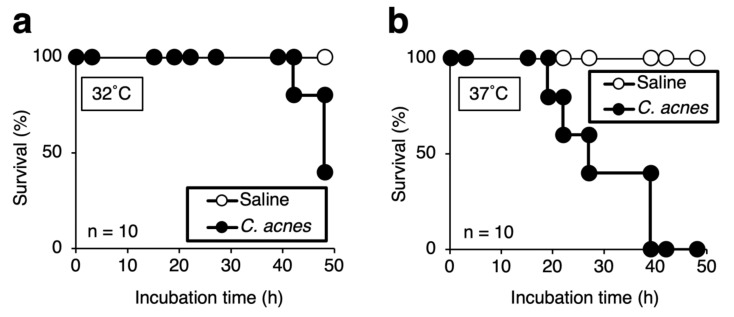
Toxic effects of *C. acnes* injected into silkworms. Silkworms were injected with saline (50 µL) or *C. acnes* cell suspension (1.6 × 10^9^ cells/50 µL) and incubated at 32 °C (**a**) and 37 °C (**b**). The number of surviving silkworms was measured for 48 h. n = 10/group.

**Figure 2 insects-12-00619-f002:**
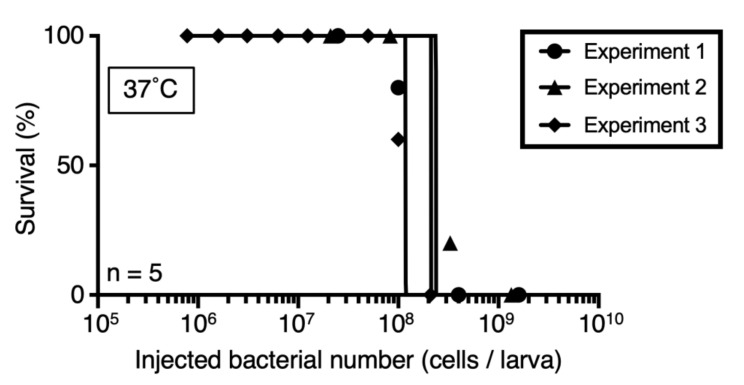
Dose response of *C. acnes* in the silkworm infection model. Silkworms were injected with saline or *C. acnes* cell suspension (Experiment 1: 2.5 × 10^7^–1.6 × 10^9^ cells/50 µL, Experiment 2: 2.1 × 10^7^–1.3 × 10^9^ cells/50 µL, or Experiment 3: 7.8 × 10^5^–4.1 × 10^8^ cells/50 µL) and incubated at 37 °C. The number of surviving silkworms was measured at 48 h after injection. n = 5/group.

**Figure 3 insects-12-00619-f003:**
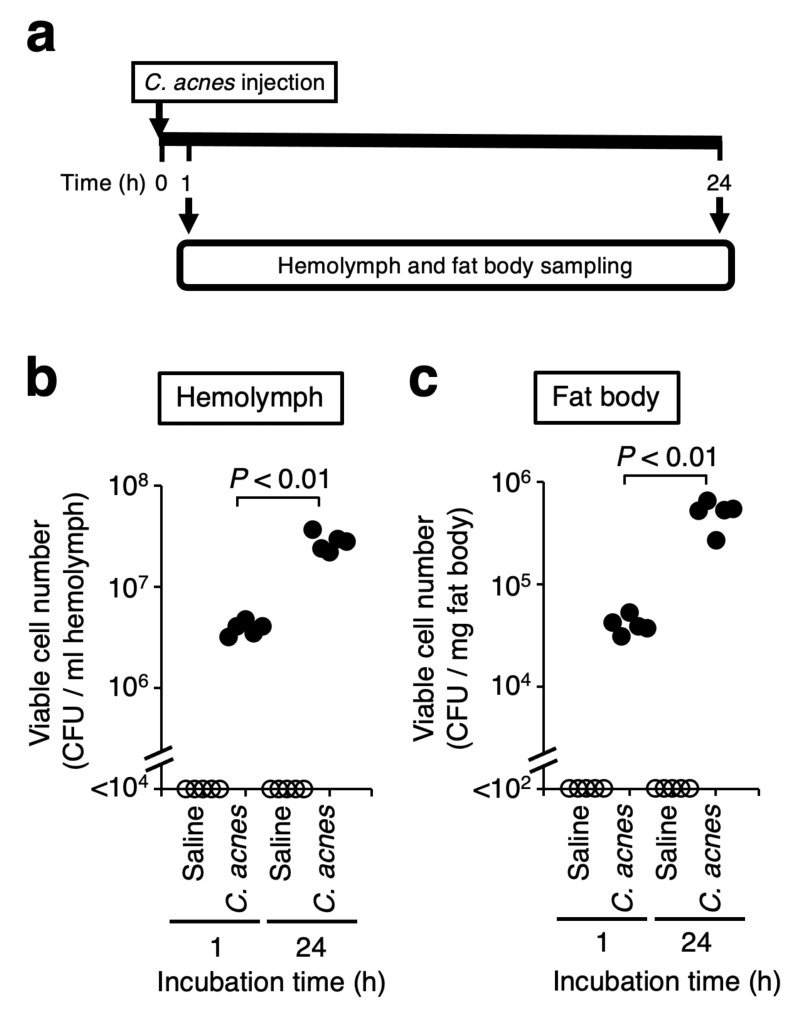
Increase in viable cell number of *C. acnes* in silkworms. (**a**) Scheme of the experiment. Silkworms were injected with saline or *C. acnes* cell suspension (3.3 × 10^8^ cells/50 µL) and incubated at 37 °C. Hemolymph (**b**) and fat body (**c**) were harvested from the silkworms at 1 or 24 h after injection. Viable cell number of *C. acnes* in the samples was measured by counting the colony-forming unit (CFU). Statistically significant differences between groups were evaluated using Student *t*-test. n = 5 / group.

**Figure 4 insects-12-00619-f004:**
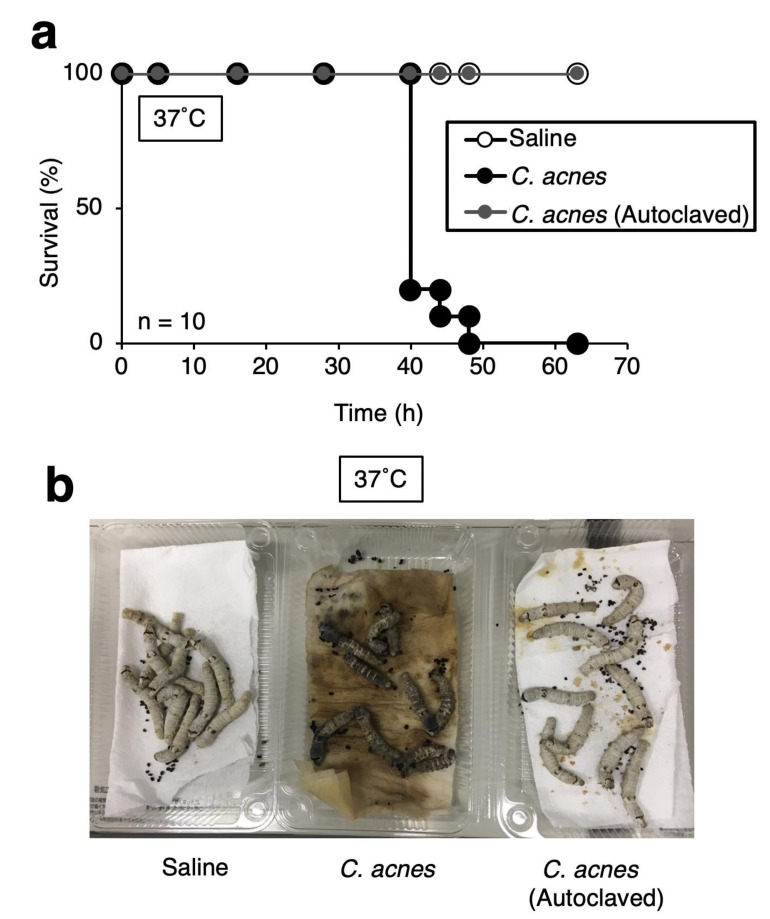
Effects of injecting autoclaved *C. acnes* cells in silkworms. (**a**) Silkworms were injected with saline (50 µL), *C. acnes* cell suspension (7.8 × 10^8^ cells/50 µL), or autoclaved *C. acnes* cell suspension and incubated at 37 °C. The number of surviving silkworms was measured for 63 h. n = 10/group. (**b**) Picture at 48 h after injection is shown.

**Figure 5 insects-12-00619-f005:**
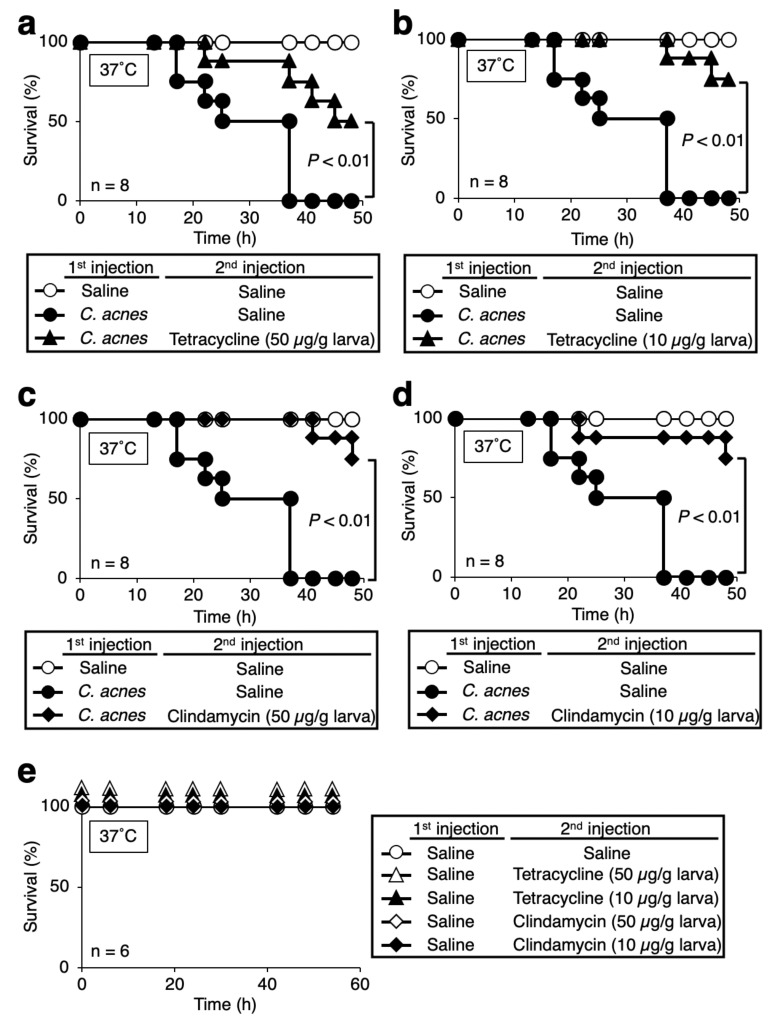
Effect of antibacterial drugs in silkworms infected by *C. acnes.* Silkworms were injected with saline (50 µL) or *C. acnes* cell suspension (3.6 × 10^8^ cells/50 µL), and then with tetracycline (50 µg/g larva) (**a**), tetracycline (10 µg/g larva) (**b**), clindamycin (50 µg/g larva) (**c**), or clindamycin (10 µg/g larva) (**d**). The number of surviving silkworms under incubation at 37 °C was measured for 48 h. Statistically significant differences between groups were evaluated using a log-rank test. n = 8/group. (**e**) Silkworms were injected with saline (50 µL), and then with tetracycline (50 µg/g larva), tetracycline (10 µg/g larva), clindamycin (50 µg/g larva), or clindamycin (10 µg/g larva). The number of surviving silkworms under incubation at 37 °C was measured for 54 h. n = 6/group.

## Data Availability

Data are contained within the article and Appendix A.

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
