# Peer review of "Evaluation of Antibacterial Drugs Using Silkworms Infected by Cutibacterium acnes"

_insects, 2021, doi:10.3390/insects12070619_

Round 1
Reviewer 1 Report
The paper deals with the development of an insect model system for the in vivo screening of candidate antibiotics active against Cutibacterium acnes. The topic is interesting and the importance of developing invertebrate models to be used in pre-clinical studies clear. However, I would suggest the authors to consider the following major points.
I feel that some controls are missing in the experiments conducted. For instance:
- Paragraph 3.2: an increase in bacterial cell count is visible both in the haemolymph and in the fat body, and this increase is likely attributable to C. acnes. However, to exclude the contribution of the indigenous microflora -which might proliferate in an unexpected way by rearing the insects at 37°C, data obtained by plating samples extracted from larvae injected with physiological solution should be added.
- Paragraph 3.4: controls injected only with 10 or 50 ug/g larva of tetracycline and clindamycin are missing. These are fundamental to exclude a possible toxic effect on the larval survival exerted by the antibiotic itself, especially considering that survival of larvae treated with 50 ug/g tetracycline (panel a in figure 5) was lower than survival of larvae treated with 10 ug/g antibiotic (panel b).
In Material and Methods section, the protocols used must be better described, as information on different aspects is missing. Here are a few examples:
- Lines 84-85:it is unclear how C. acnes cells, once harvested by the GAM plate, were quantified. Similarly, for how long and at which temperature were C. acnes plates incubated?
- Line 90-91: larvae survival was monitored “over time”: please, specify for how long.
- Moreover, is it rather unclear why the amounts of cells injected in the experiments were variable (for example, 3.3x10^8, according to LD50, for proliferation into haemolymph and fat body, 7.8x10^8 for comparing alive and autoclaved C. acnes cells, etc)
Minor points:
- Names of microbial strains should be in italic (lines 138, 140 etc)
- Line 160: this sentence is unclear.
- Paragraph 3.4: I would briefly explain here -rather than in the final discussion- why these antibiotics and at these specific concentrations were tested
- Lines 213: sentence unclear, to be rephrased.
Overall, the manuscript would benefit from an English revision.
Author Response
Reviewer 1
The paper deals with the development of an insect model system for the in vivo screening of candidate antibiotics active against Cutibacterium acnes. The topic is interesting and the importance of developing invertebrate models to be used in pre-clinical studies clear. However, I would suggest the authors to consider the following major points.
I feel that some controls are missing in the experiments conducted. For instance:
- Paragraph 3.2: an increase in bacterial cell count is visible both in the haemolymph and in the fat body, and this increase is likely attributable to C. acnes. However, to exclude the contribution of the indigenous microflora -which might proliferate in an unexpected way by rearing the insects at 37°C, data obtained by plating samples extracted from larvae injected with physiological solution should be added.
According to the reviewer’s comment, we added the data of control, which is a sample extracted from silkworms injected with a saline in Figure 3b and c of the revised manuscript.
- Paragraph 3.4: controls injected only with 10 or 50 ug/g larva of tetracycline and clindamycin are missing. These are fundamental to exclude a possible toxic effect on the larval survival exerted by the antibiotic itself, especially considering that survival of larvae treated with 50 ug/g tetracycline (panel a in figure 5) was lower than survival of larvae treated with 10 ug/g antibiotic (panel b).
According to the reviewer’s comment, we examined and added data in Figure 5e of the revised manuscript.
In Material and Methods section, the protocols used must be better described, as information on different aspects is missing. Here are a few examples:
- Lines 84-85:it is unclear how C. acnes cells, once harvested by the GAM plate, were quantified. Similarly, for how long and at which temperature were C. acnes plates incubated?
According to the reviewer’s comment, we described the method for harvesting C. acnes cells grown on the GAM agar plate in the revised manuscript (Page 2, line 74-82).
Page 2, lines 74-79.
Physiologic saline (2 ml) was added to the GAM agar plate and suspended using a cell spreader. The C. acnes cell suspension was collected from the agar plate and measured absorbance at 600 nm using a spectrophotometer (U-1100; Hitachi Ltd., Tokyo, Japan). The C. acnes cells were suspended in physiologic saline to 5-10 of absorbance at 600 nm, and the cell suspension was used in the experiments.
In addition, we described the incubating temperature and time in the revised manuscript (Page 2, lines 73-74).
Page 2, line 73-74.
The C. acnes ATCC6919 strain was grown on GAM agar plates at 37ËšC for 3 days under anaerobic conditions.
- Line 90-91: larvae survival was monitored “over time”: please, specify for how long.
Following the reviewer’s comment, we described the monitoring time in legends of Figures 1, 4, and 5 in the revised manuscript (Page 4, line 157, page 6, lines 189, page 7, lines 204).
Page 4, line 157.
The number of surviving silkworms was measured for 48 h.
Page 6, line 189.
The number of surviving silkworms was measured for 63 h.
Page 7, line 204.
The number of surviving silkworms was measured for 48 h.
- Moreover, is it rather unclear why the amounts of cells injected in the experiments were variable (for example, 3.3x10^8, according to LD50, for proliferation into haemolymph and fat body, 7.8x10^8 for comparing alive and autoclaved C. acnes cells, etc)
In this study, we estimated injected cell numbers by counting the colony-forming unit of the bacterial suspension in each experiment. Therefore, the amounts of injected cell numbers were variable in each experiment. According to the reviewer’s comment, we described the method for the determination of the amounts of injected cells in the revised manuscript (Page 2, line 79-82).
Page 2, lines 79-82.
For calculating the cell number in the cell suspension, the suspension of C. acnes cells was 107-fold diluted with saline, and a 100 μl aliquot was spread on GAM agar plates. After incubation at 37°C for 3 days under anaerobic conditions, the number of colonies was counted for determining the colony-forming unit (CFU).
Minor points:
- Names of microbial strains should be in italic (lines 138, 140 etc)
Thank you very much for the comment. Following the reviewer’s comments, we revised the words in the revised manuscript.
Page 4, line 142.
Aspergillus fumigatus
Page 4, line 144, line 146, 152, 153, 155, 159, page 5, line 165, 168, 169, 171, 172, 173, 182, 183, 184, 187, 188, 193, 197, and 199.
C. acnes
Page 5 line 177.
Porphyromonas gingivalis
Page 5, line 178.
P. gingivalis
- Line 160: this sentence is unclear.
According to the reviewer’s comment, we changed the sentence in the revised manuscript (Page 5, line 164-165).
Page 5, line 164-165.
We next investigated whether C. acnes cells can grow in the silkworm body. The scheme of the experiment is shown in Figure 3a.
- Paragraph 3.4: I would briefly explain here -rather than in the final discussion- why these antibiotics and at these specific concentrations were tested
Following the reviewer’s comment, we added the sentence in the result section of the revised manuscript (Page 6, line 193-196).
Page 6, line 193-196.
In clinical, tetracycline and clindamycin are administered 250-500 mg/person/day and 600 mg/person/day, respectively [30-32]. Based on the information, we determined injection amounts of the antibacterial drugs in the experiments.
- Lines 213: sentence unclear, to be rephrased.
Following the reviewer’s comment, we changed the sentence in the Discussion section of the revised manuscript (Page 8, lines 229-230).
Page 8, line 229-230.
The results suggest that C. acnes causes bacteremia by growth in the silkworm body.
Overall, the manuscript would benefit from an English revision.
Following the referee’s comment, the revised manuscript was edited by professional native English-speaking science editors (SciTechEdit International, LLC, CO, USA).

Reviewer 2 Report
I have only one minor question below.
1. Line 179-181: The authors claimed that “Injection of C. acnes cells killed silkworms, but injection of autoclaved C. acnes cells did not (Figure 4). These results suggest that the pathogenicity of C. acnes in silkworms depends on the viability of the C. acnes cells.” However, I was confused by the digital photos of silkworms in the figure 4b where the silkworms injected by the autoclaved C. acnes are looked sick as there are distinct aqueous brown silkworm excreta on the white tissue paper like the phenomenon happened in the silkworms infected by the live C. acnes. Thus, whether will these silkworms injected by the autoclaved C. acnes cell suspension die in the end? If so, the conclusion made by the author that pathogenicity of C. acnes in silkworms depends on the viability of the C. acnes cells should be revised accordingly.
Author Response
Reviewer 2
- Line 179-181: The authors claimed that “Injection of C. acnes cells killed silkworms, but injection of autoclaved C. acnes cells did not (Figure 4). These results suggest that the pathogenicity of C. acnes in silkworms depends on the viability of the C. acnes cells.” However, I was confused by the digital photos of silkworms in the figure 4b where the silkworms injected by the autoclaved C. acnes are looked sick as there are distinct aqueous brown silkworm excreta on the white tissue paper like the phenomenon happened in the silkworms infected by the live C. acnes. Thus, whether will these silkworms injected by the autoclaved C. acnes cell suspension die in the end? If so, the conclusion made by the author that pathogenicity of C. acnes in silkworms depends on the viability of the C. acnes cells should be revised accordingly.
In the experiment, the silkworms are incubated at 37ËšC without food, and they die after 4 days even in the saline group. We confirmed that the silkworm injected with the autoclaved C. acnes cells does not die until 62 hours under the conditions of this experiment. On the other hand, as the reviewer pointed out, the silkworms injected with the autoclaved C. acnes cells seem to be not healthy compared with those in the saline group. We assumed the reason is that the hemolymph of silkworms was melanized and the melanization might affect the silkworm condition. According to the reviewer’s comment, we changed the sentence in the Result section and added the sentences in the Discussion section of the revised manuscript (Page 6, lines 184-185, page 8, lines 217-220).
Page 6, line 184-185.
These results suggest that the heat-stable components of C. acnes don’t cause acute silkworm death in the experimental condition.
Page 8, line 217-220.
The silkworms injected with the autoclaved C. acnes cells seem to be not healthy compared with those in the saline group (Figure 4b). We assumed the reason is that the hemolymph of silkworms was melanized and the melanization might affect the silkworm condition.

Reviewer 3 Report
This manuscript describes the work for establishing a silkworm model of C. acnes infection and evaluated antibacterial drugs treatment of the infected host. The experiment design was in general direct and clear, and the results were well put.
1. which silkworm strain did the authors use for the experiment? On which day of fifth instar larvae were the bacteria injected?Please specify.
2. What are the main symptons after silkworm infection with C. acnes, please describe more in details.
3. why the authors choose to inject the bacteria into the silkworm, instead of feeding orally such as as feed additive? Especially in the antibacterial drug experiment, the silkworm were injected twice. Injection without proper disinfection would cause death of silkworm due to decreased host immunity.
4. In methods 2.5, "the number of connies was counted and the number of viable cells in the sample was calculated", but the authors only presented the number of viable cells without the colony numbers? How did the authors determined whether the cells are viable? Describe in detail in the methods.
5. Check the font throughout, for example, Line138, the latin name should be italicized. Line 140, C. acnes should be italicized.
Author Response
Reviewer 3
This manuscript describes the work for establishing a silkworm model of C. acnes infection and evaluated antibacterial drugs treatment of the infected host. The experiment design was in general direct and clear, and the results were well put.
1. which silkworm strain did the authors use for the experiment? On which day of fifth instar larvae were the bacteria injected?Please specify.
In this study, we used the silkworm strain, Hu・Yo×Tukuba・Ne. According to the reviewer’s comment, we added the information in the revised manuscript (Page 2, line 85).
For the infection experiments, we used the fifth-instar larvae that were fed with the artificial diet overnight. According to the reviewer’s comment, we added the information in the revised manuscript (Page 2, lines 91-92).
Page 2, lines 91-92.
Silkworm fifth instar larvae were fed an artificial diet (1.5 g; Silkmate 2S; Ehime-Sanshu Co., Ltd., Ehime, Japan) overnight.
- What are the main symptons after silkworm infection with C. acnes, please describe more in details.
We assumed that the silkworms infected with C. acnes cause sepsis by the proliferation of C. acnes cells in the hemolymph and on tissues of the silkworms. The C. acnes cells grow in the hemolymph and on the tissues of the silkworms. C. acnes use the nutrition of the silkworms and might cause toxicity to the tissues. The melanization of the hemolymph of silkworms caused by C. acnes might affect tissue toxicity. According to the reviewer’s comment, we added the sentences in the Discussion section of the revised manuscript (Page 8, line 220-225).
Page 8, line 220-225.
The C. acnes cells grow in the hemolymph and on the tissues of the silkworms. C. acnes use the nutrition of the silkworms and might cause toxicity to the tissues. The melanization of the hemolymph of silkworms caused by C. acnesmight affect tissue toxicity. We assumed that the silkworms infected with C. acnes cause sepsis by the proliferation of C. acnes cells in the hemolymph and on tissues of the silkworms.
- why the authors choose to inject the bacteria into the silkworm, instead of feeding orally such as as feed additive? Especially in the antibacterial drug experiment, the silkworm were injected twice. Injection without proper disinfection would cause death of silkworm due to decreased host immunity.
Because C. acnes causes bloodstream infections, we selected injection the C. acnes into the hemolymph of the silkworms.We confirmed that the silkworm does not die by injection twice in the experiments (Figure 5e).
According to the reviewer’s comment, we added the sentences in the Discussion section of the revised manuscript (Page 8, lines 213-214).
Page 8, lines 213-214.
Because C. acnes causes bloodstream infections, we selected injection the C. acnes into the hemolymph of the silkworms.
- In methods 2.5, "the number of connies was counted and the number of viable cells in the sample was calculated", but the authors only presented the number of viable cells without the colony numbers? How did the authors determined whether the cells are viable? Describe in detail in the methods.
In this study, we determined viable cell numbers by counting the colony-forming unit of the silkworm samples. Viable cells of C. acnes in samples form colonies on GAM agar plates after incubation at 37ËšC for 3 days under anaerobic conditions. Therefore, the colony number on the GAM agar plates corresponds to the viable cell number in the samples. According to the reviewer’s comment, we described the method in detail in the revised manuscript (Page 3, line 100-111).
Page 3, lines 100-111.
For calculating the C. acnes cell number in the hemolymph, the hemolymph was 103 or 104-fold diluted with saline, and a 100 μl aliquot was spread on GAM agar plates. The number of colonies was counted after incubation at 37ËšC for 3 daysunder anaerobic conditions. The viable cell number of C. acnes in the hemolymph was calculated as the colony-forming unit (CFU) per 1 ml.
Fat body isolation was performed as previously described [26]. Silkworms were placed on ice for 15 min. The fat body was isolated from the dorsolateral region of each silkworm and rinsed in saline. The wet weight of the fat body was measured. The fat body was homogenized in 1 ml of saline, and the lysate was obtained. The lysate was 102 or 103-diluted with saline, and a 100 μl aliquot was spread on GAM agar plates. The number of colonies was counted after incubation at 37ËšC for 3 days under anaerobic conditions. The viable cell number of C. acnes per wet weight of the fat body was determined.
- Check the font throughout, for example, Line138, the latin name should be italicized. Line 140, C. acnes should be italicized.
Thank you very much for the comment. Following the reviewer’s comments, we revised the words in the revised manuscript.
Page 4, line 142.
Aspergillus fumigatus
Page 4, line 144, line 146, 152, 153, 155, 159, page 5, line 165, 168, 169, 171, 172, 173, 182, 183, 184, 187, 188, 193, 197, and 199.
C. acnes
Page 5 line 177.
Porphyromonas gingivalis
Page 5, line 178.
P. gingivalis

Round 2
Reviewer 1 Report
All points raised during the previous revision have been addressed. I think that the manuscript in the present form could be considered for publication.